# An Analysis of Slope Stability in the Penipe–Baños Road by Applying Empirical Methods, Kinematic Analysis and Remote Photogrammetry Techniques

**Luis Jordá-Bordehore** [1,*], **Lissette G. Albán** [2], **Ramiro C. Valenzuela** [2], **Gianella Bravo** [2], **Melanie Menoscal-Menoscal** [2], **Erwin Larreta** [2], **Daniel Garces** [2] **and Maurizio Mulas** [2]

1 Department of Engineering and Surface Morphology, Polytechnic University of Madrid, 28040 Madrid, Spain
2 Faculty of Engineering in Earth Sciences (FICT), ESPOL Polytechnic University, Gustavo Galindo Campus, Guayaquil P.O. Box 09-01-5863, Ecuador; lisagome@espol.edu.ec (L.G.A.); rvalenzu@espol.edu.ec (R.C.V.); gmbravo@espol.edu.ec (G.B.); mmenosca@espol.edu.ec (M.M.-M.); elarreta@espol.edu.ec (E.L.); ogarces@espol.edu.ec (D.G.); mmulas@espol.edu.ec (M.M.)
* Correspondence: l.jorda@upm.es

**Abstract:** The purpose of this work is to analyze the stability of four slopes along the Penipe–Baños road, which is situated in the provinces of Chimborazo and Tungurahua and where there are occasionally rockfalls that hinder passage and endanger road users. The methodology used to conduct the analysis was based on data collection with the help of remote techniques such as structure from motion, which allows us to obtain slope data using photogrammetry. Empirical methods such as slope mass rating, Q-slope, the kinematic method and the Rockfall Hazard Rating System method were used. These methods were evaluated with Rocfall3 software for the analysis of the fall trajectory of rock blocks. The results of this work show that the slopes studied do not represent a greater risk to the road than other slopes close to those studied, but these could not be analyzed due to their lack of accessibility and the danger of obtaining data under those conditions. The study of these different methods demonstrates the reliability of low-cost, remote techniques in the facilitation of analysis in places with similar conditions.

**Keywords:** slope stability; geomechanical stations; photogrammetry; RHRS; rockfall hazard

## 1. Introduction

The Cahuají–Pillate–Cotaló road was inaugurated on 18 June 2015. This road has a length of 26 km and a width of 11.50 m. It was built as an alternative route to the old Penipe–Baños road, which has been affected since 1999 by different eruptions of the Tungurahua volcano. The old road is crossed by more than seven ravines, through which mud and lahar flows descend as a result of the volcano eruptions. Due to the high risk in the area, the risk management secretary has declared it an emergency zone on several occasions. This road connects the cantons of Baños and Penipe, which are agricultural and tourist cantons belonging to the Tungurahua and Chimborazo provinces, respectively.

The study area is in a mountainous zone with natural and artificial slopes composed of volcanic material. At present, the slopes have frequent rockslides, constituting a risk to road users and residents of the area.

Regarding the geology of the site, the slopes are on the Mulmui and Igualata volcanoes (Pliocene), which are extinct volcanoes. On this site, outcrops mainly include pyroclastic materials from fine-grained tuff to coarse tuff pumice, as well as lavas of andesitic composition [1].

The goal of this research is to analyze the stability of four slopes located on the Cahuají–Pillate–Cotaló road, which is an alternate route to the Penipe–Los Pájaros (Baños) road (Figure 1), by applying different approaches. The methodologies include SfM (structure

from motion) photogrammetry as well as the empirical methods Q-slope, RMR (rock mass rating), SMR (slope mass rating), limit equilibrium–kinematic analysis (DIPS), and the RHRS (Rockfall Hazard Rating System) [2]. Initially, a finite element study was considered based on rock mass parameters (Hoek and Brown criteria) and the global geometry of each slope, but clear kinematic failure modes were identified and no evidence was found on slope instabilities through the rock mass. Neither global nor cyclical failure were observed in the area; therefore, no further analysis was considered to be required.

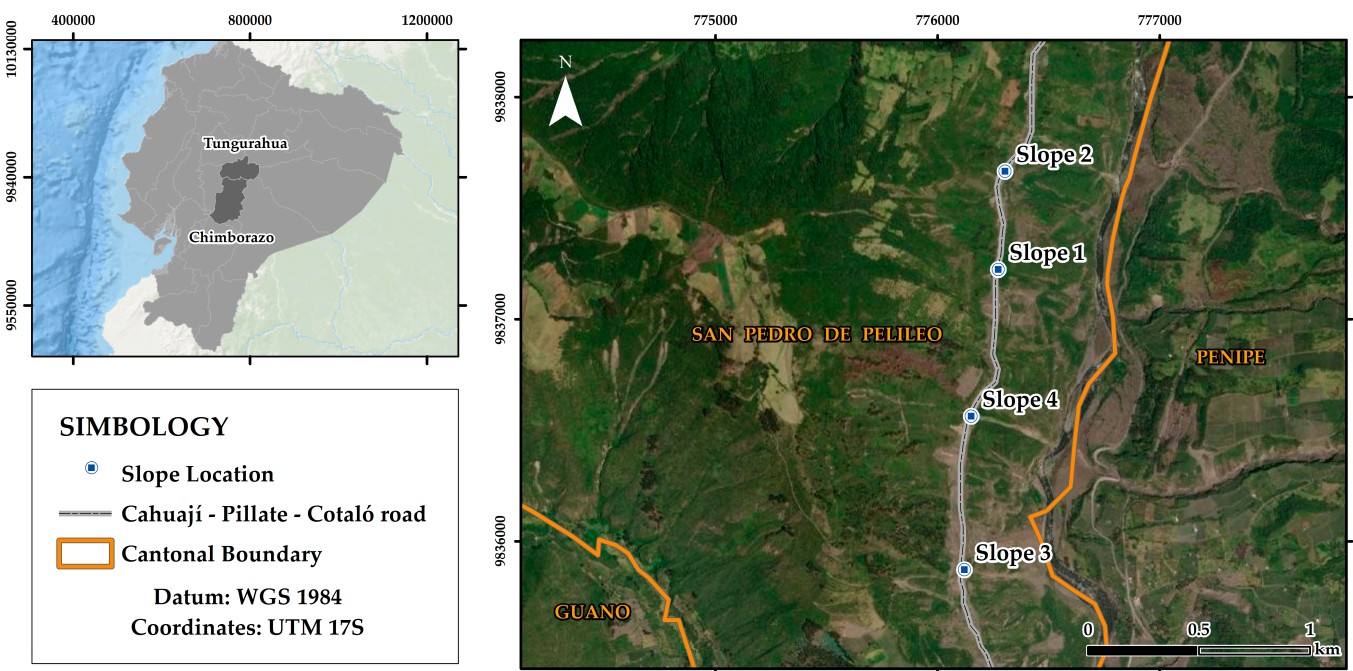

**Figure 1.** The location of the study area at the Cahují–Pillate–Cotaló road, Tungurahua and Chimborazo provinces, Ecuador.

A geological–geotechnical survey (geomechanical stations) was carried out to determine the stability of the four slopes through the application of empirical methods [3], such as slope mass rating [4] and Q-slope [5] as well as kinematic analysis. These geomechanical stations have been analyzed in conjunction with photogrammetry and field measurements obtained from the few accessible places on the slopes [6]. These techniques and methodologies were completed with the Rockfall Hazard Rating System (RHRS) method, developed by Pierson et al. [7], in order to characterize the stability of the rock slopes and compare data with those obtained with Rocfall software to evaluate the trajectories of the rockfalls.

## 2. Materials and Methods

Geomechanical stations were used to collect geotechnical data from the slopes. A geomechanical station is a set of rock mechanics observations of the properties of the rock matrix and rocky massif [8]. Among other parameters, uniaxial compressive strength (using a sclerometer), the orientation of fracture families (compass) and the features of joints or discontinuities (roughness, filling, spacing), as well as the rock quality designation, were obtained.

To complement data collection and to better characterize the inaccessible parts of the slopes, remote techniques such as SfM (structure from motion) photogrammetry were used [9,10]. The data obtained were used for the application of empirical methods such as SMR, Q-slope (where high scores mean stability), and the Rockfall Hazard Rating System (RHRS), which gives a score to the slopes according to the data and characteristics of the site, with the highest score representing the most unfavorable slope. In addition, kinematic analysis was also implemented. The studied slopes are shown in Figure 2.

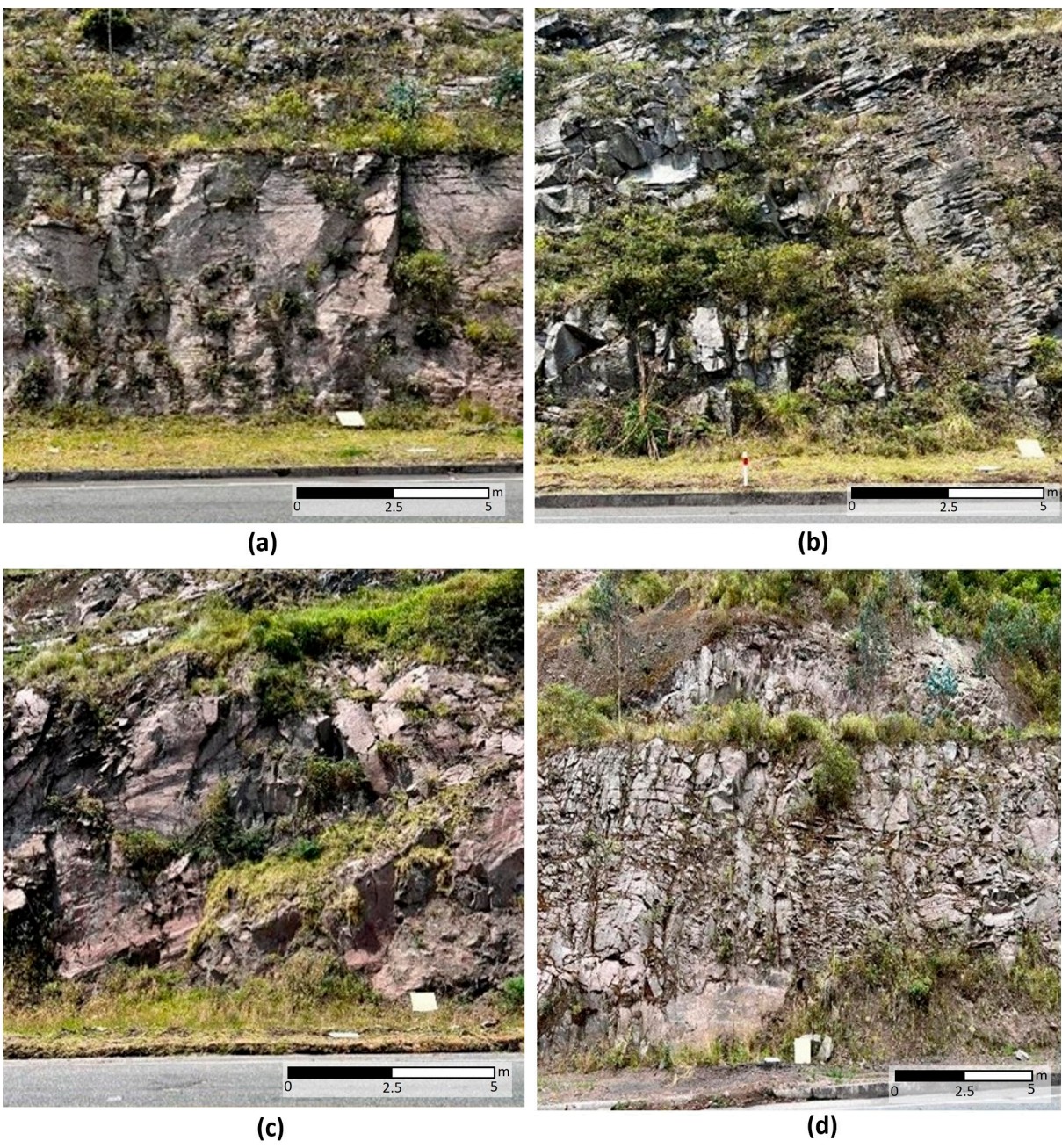

**Figure 2.** Studied slopes at the Cahuají—Pillate—Cotaló road (alternate route to the old Baños—Penipe road): (**a**) Slope 1; (**b**) Slope 2; (**c**) Slope 3; (**d**) Slope 4.

### 2.1. Photogrammetry SfM (Structure from Motion)

Field data collection is essential for conducting a realistic analysis; however, in many instances, gathering geometric and geotechnical data in areas with high escarpments and rockfall risks can be perilous. The collection of such data is often restricted by the location and accessibility of the studied slope during manual measurements, as well as the availability of equipment, such as topographic tools, necessary for accurately obtaining the desired data.

In this scenario, a remote technique approach using photogrammetry, specifically structure from motion (SfM), was employed, due to the inherent dangers associated with collecting data below the slopes. The SfM (structure from motion) technique is recognized as a high-resolution and low-cost photogrammetry method sharing principles with stereoscopic photogrammetry, where a 3D point cloud is generated by overlaying images [11–14].



For data collection, a cell phone camera, the iPhone 13 Pro, was utilized. Multiple photos were captured of each of the four slopes (Slope 1: 188 photos, Slope 2: 260 photos, Slope 3: 165 photos, Slope 4: 133 photos). To establish orientation in photogrammetry, a 60 cm × 40 cm board served as a reference plane, which was placed horizontally with one side oriented towards the geographic north to use the three corners of the board as control or reference points to orient the model. Measurements were also taken in the field using a Brunton compass and a flexometer. To determine the slope height, a BOSCH laser meter was employed, utilizing a laser to measure the height accurately.

Subsequently, the photos were processed in Agisoft software (2016) [11] to create the point cloud for each slope. Finally, to verify the results, the dip and dip direction measurements of the reference plane used were compared with the results of the same measurements in the CloudCompare software version 2.12 [12].

### 2.2. Geomechanical Stations

Geomechanical stations play a crucial role in acquiring pertinent slope data, which are grounded in existing site conditions and discontinuity behavior. A geomechanical station can be defined as a point or observation area of a rock outcrop, encompassing an approximate environment of about five meters where the discontinuities and rock matrix are characterized, extracting physical and mechanical parameters [13]. The analysis includes an examination of the distribution, orientation, and dip of the discontinuities, as well as the conditions of alteration and resistance of the walls in the rock matrix and lips of discontinuities, in order to define the families influencing the massif. Therefore, these data were correlated with the data later obtained with SfM, incorporating an automatic analysis of the discontinuities with the free-to-use software Discontinuity Set Extractor (DSE) in the MATLAB environment. This analysis allowed us to identify the discontinuities from the point cloud made previously [14]. To conduct each geomechanical station, a Brunton compass model 9077, a PCE—HT 225, a Schmidt original sclerometer, a geomechanical notebook, and a BOSCH GLM400C laser meter, among other basic tools, were employed. The features obtained from the geomechanical stations enabled the application of various slope classification methods, which are among several approaches used in contemporary slope analysis [15–19].

### 2.3. SMR (Slope Mass Rating)

The geometrical classification known as SMR has its origin in the refinement of the joint and slope orientation of the basic RMR (rock mass rating), which was developed by Bieniawski [20]. The adjustment factors to derive the SMR value from the mentioned RMR depend on the geometrical relationship between slope orientations and discontinuities, as well as the excavation method used. This classification is employed to evaluate the stability of rock slopes.

The SMR is obtained from this equation [4]:

$$SMR = RMR_b + (F1 \times F2 \times F3) + F4 \tag{1}$$

Here, $RMR_b$ represents the value of the basic rock mass rating, F1 is the factor depending on the angle between the direction of the joints and the slope face, F2 is the factor depending on the dip of the joint at the plane rupture, F3 is the factor reflecting the relationship between joint and slope dips and F4 is the factor dependent on the method employed for slope excavation.

### 2.4. Q-Slope

Q-slope rock mass classification [21] is an empirical method used to assess the behavior of rock slopes excavated in the field, applicable to both road construction and surface mining. This method is derived from the Q index [22], extensively utilized for characterizing and evaluating rock masses in subway projects. The parameters maintained for Q-slope rock mass are RQD, Jn, Jr and Ja. Additional parameters such as Jwice and SRF (strength reduction factor) are determined based on various conditions observed over the course of slope exposure, as detailed in the following tables (Tables 1–3). For the specific applications of SRFa, SRFb and SRFc we refer the reader to the original criteria references [21]. For the application of SRF, the highest score among SRFa, SRFb and SRFc should be used.

**Table 1.** Environmental and geological condition numbers (Jwice) [21].

| Description | Desert Environment | Wet Environment | Tropical Storms | Ice Wedging |
|---|---|---|---|---|
| Stable structure; competent rock | 1.0 | 0.7 | 0.5 | 0.9 |
| Stable structure; incompetent rock | 0.7 | 0.6 | 0.3 | 0.5 |
| Unstable structure; incompetent rock | 0.8 | 0.5 | 0.1 | 0.3 |
| Unstable structure; incompetent rock | 0.5 | 0.3 | 0.05 | 0.2 |

**Table 2.** Strength reduction factor maxima for $SRF_a$ [21].

| Description | $SRF_a$ |
|---|---|
| Slight loosening due to surface location disturbance from blasting or excavation | 2.5 |
| *Loose blacks, tension cracks, joint shearing, weathering, susceptibility, severe blasting disturbance | 5 |
| As above (*), but with strong susceptibility to weathering | 10 |
| Slope is in the advanced stage of erosion and loosening due to erosion by water and/or ice-wedging effects | 15 |

**Table 3.** Strength reduction factor maxima for $SRF_b$ [21].

| Description | $\sigma_c/\sigma_1$ | $SRF_b$ |
|---|---|---|
| Moderate stress–strength range | 50–200 | 2.5–1 |
| High stress–strength range | 10–50 | 5–2.5 |
| Localized intact rock failure | 5–10 | 10–5 |
| Crushing or plastic yield | 2.5–5 | 15–10 |
| Plastic flow of strain softened material | 1–2.5 | 20–15 |

Q-slope is calculated using this equation:

$$Q_{slope} = \frac{RQD}{J_n} \times \left(\frac{J_r}{J_a}\right)_O \times \frac{Jwice}{SRF_{slope}} \tag{2}$$

where RQD is the rock quality designation obtained using the joints-per-meter criteria, $J_n$ is the diaclase index value, $J_r$ is the roughness index of discontinuities value, $J_a$ is the index discontinuities alteration value, O is the discontinuity orientation factor, Jwice is the value of the environmental and geological conditions and $SRF_{slope}$ is the strength reduction factor.

Once the value of Q-slope is obtained, the value of the maximum slope angle can be determined using the following equation [21]:

$$\beta = 20\log_{10}Q_{slope} + 65 \tag{3}$$

where β is the maximum slope angle.

*2.5. RHRS (Rockfall Hazard Rating System)*

The Rockfall Rating System was designed to identify slopes with high risk requiring immediate attention or more comprehensive study [7]. However, in many mountainous areas, rock cuts may need excavation and, due to outdated practices at the time of this methodology's development, poor blasting techniques and aggressive ripping have resulted in slopes more susceptible to rockfall detachment.

To implement this methodology, it is essential to develop an inventory of the slopes and conduct a preliminary classification of the slopes into three categories, A, B and C (high, moderate or low), based on the level of threat. The goal is to document locations where rockfalls occur and identify hazardous slopes. With detailed classification, the aim is to numerically differentiate the risk in the places where landslide may occur. Based on the score, planning and organizing interventions can be prioritized, with higher scores indicating greater risk.

Maintaining photographic records of slopes is crucial as they allow criteria to be correlated with actual site conditions. This classification is a gradual process designed to identify dangerous slopes and determine the necessary steps for mitigation and corrections.

For the application of the modified RHRS, studies in southern Italy revealed that the RHRS method was susceptible to certain categories of the original methodology and these data varied depending on each evaluator's perspective.

The RHRSmod [23] incorporates ratings for slope height, trench effectiveness, average vehicle risk, visual decision percentage, roadway width, the geological characteristics of the site, rockfall volume and block size, weather, presence of water on the slope and rockfall history. Importantly, this method includes the SMR category, obtained previously in this study, demonstrating its simplicity and objectivity in slope classification.

*2.6. Kinematic Analysis Using the Limit Equilibrium and Discontinuity Method*

Kinematic analysis is a method employed to evaluate the stability of a rock slope through relevant information such as the identification of the slope's discontinuities, the dip and dip direction of the discontinuities, the dip and dip direction of the slope and the friction coefficient of the rock composing the slope, among other parameters.

This kinematic analysis was conducted graphically using Rocscience's DIPS_v8 software. Through the stereographic projections used by the program, areas with potential faults indicating instability were identified if a discontinuity was projected in a specific region. Conversely, the slope was deemed stable in the absence of such discontinuities.

In order to assess the kinematic stability of the rock blocks, determining the shear strength of discontinuities is necessary. In this study, Barton's (2002) criteria for the "frictional component" of joints were applied, with cohesion set to 0 for conservative design considerations. The "frictional component" (FC) was calculated using the following equation [24]:

$$FC \; (deg) = \tan^{-1}\left(\frac{J_r}{J_a} \times J_w\right) \tag{4}$$

where $J_r$ is the value of the roughness index of discontinuities, $J_a$ is the value of index discontinuity alteration and $J_w$ is the value of the water presence index.

**3. Results**

Discontinuity Set Extractor (DSE) software was used to compare the measured results obtained in the field. Using the program's results, it is possible to determine the discontinuities of each of the slopes through the graphics provided by this software in conjunction with CloudCompare. The identified discontinuities serve as the basis for all the parameters used in the calculation methods employed for this study. Figure 3 shows the main discontinuities considered for each slope:

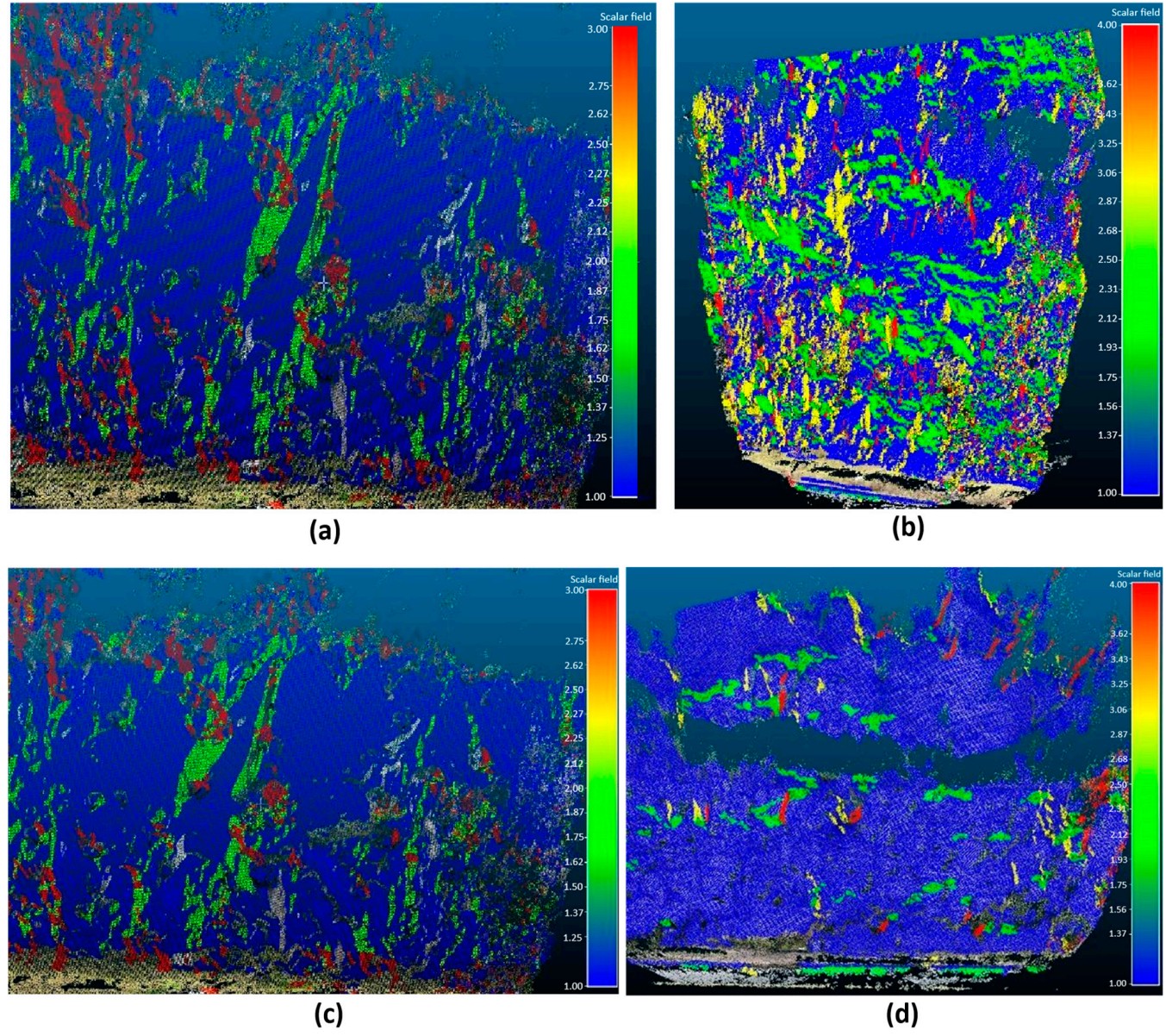

**Figure 3.** Discontinuities identified with DSE: (**a**) Slope 1; (**b**) Slope 2; (**c**) Slope 3; (**d**) Slope 4.

Table 4 shows the details of the discontinuities identified in each slope:

**Table 4.** Description of discontinuities found in each of the slopes.

| Joint Sets | Slope | | | |
|---|---|---|---|---|
| | **1** | **2** | **3** | **4** |
| Joint 1 DipDir/Dip | 057/87 | 320/86 | 097/064 | 292/88 |
| Joint 2 DipDir/Dip | 036/53 | 072/70 | 194/83 | 322/86 |
| Joint 3 DipDir/Dip | N.I | 176/86 | 354/77 | 052/86 |

### 3.1. Slope Mass Rating (SMR)

The results of the basic RMR obtained indicate that the rock in each of the slopes falls within the range of 50 to 70. When applying the correction factors for each of the failure cases, Planar (P) and Toppling (T) (based on measurements and observations, with no potential wedge failure identified), it was found that the first two slopes were partially stable, the third was stable and the fourth was considered unstable. This instability is

primarily attributed to the orientation of the discontinuities relative to the slope. The following table shows the parameters used and the results obtained for each of the slopes (Table 5).

**Table 5.** Characteristic geomechanical parameters of the RMR and SMR in the study area.

| Parameters | | Slope | | | |
|---|---|---|---|---|---|
| | | 1 | 2 | 3 | 4 |
| Lithology | | Ignimbrite | Ignimbrite | Ignimbrite | Ignimbrite |
| Slope height (m) | | 10.80 | 23.70 | 8.00 | 11.15 |
| UCS (MPa) | | 45.08 | 33.32 | 35.28 | 42.14 |
| RQD (%) | | 93 | 94 | 97 | 85 |
| Joint spacing value | | 3 | 3 | 3 | 3 |
| Joint condition value | | 2 | 2 | 2 | 2 |
| Presence of water value | | 15 | 15 | 15 | 15 |
| RMR basic value | | 67.30 | 68 | 68 | 57 |
| F1 P | J1 | 0.40 | 0.15 | 0.40 | 0.15 |
| | J2 | 0.15 | 0.15 | 0.15 | 0.15 |
| | J3 | - | 0.15 | 0.15 | 0.15 |
| F2 P | J1 | 1.00 | 1.00 | 1.00 | 1.00 |
| | J2 | 1.00 | 1.00 | 1.00 | 1.00 |
| | J3 | - | 1.00 | 1.00 | 1.00 |
| F3 P | J1 | −6.00 | 0.00 | −6.00 | 0.00 |
| | J2 | −60.00 | −6.00 | 0.00 | 0.00 |
| | J3 | - | 0.00 | 0.00 | 0.00 |
| F1 T | J1 | 0.15 | 1.00 | 0.15 | 1.00 |
| | J2 | 0.15 | 0.15 | 0.15 | 0.40 |
| | J3 | - | 0.15 | 0.15 | 0.15 |
| F2 T | J1 | 1.00 | 1.00 | 1.00 | 1.00 |
| | J2 | 1.00 | 1.00 | 1.00 | 1.00 |
| | J3 | - | 1.00 | 1.00 | 1.00 |
| F3 T | J1 | −25.00 | −25.00 | −25.00 | −25.00 |
| | J2 | −25.00 | −25.00 | −25.00 | −25.00 |
| | J3 | - | −25.00 | −25.00 | −25.00 |
| Excavation | | Blasting or excavation | Blasting or excavation | Blasting or excavation | Blasting or excavation |
| F4 | | 0.00 | 0.00 | 0.00 | 0.00 |
| Set of joints | | J2 | J1 | J1 | J1 |
| SMR | | 58.30 | 43.00 | 64.25 | 32.00 |
| Stability | | Partially stable | Partially stable | Stable | Unstable |
| Future case | | P | T | T | T |

*3.2. Q-Slope*

The first slope was identified as unstable, while the remaining three slopes were deemed stable. This distinction arises from the Q-slope method, where slope inclination is a critical factor. If the inclination is less than the calculated angle (β), the slope is classified as stable. This consideration leads to the results indicating stability for slopes 2, 3 and 4, as they have a low inclination that qualifies them as stable in this method (Table 6) (Figure 4).

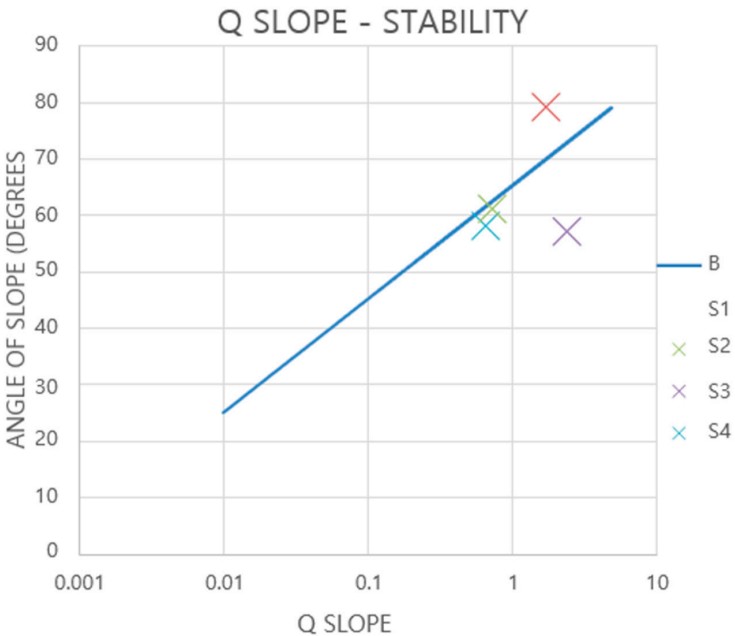

**Figure 4.** Q-slope.

**Table 6.** Parameters of the geomechanical characteristics of the Q-slope index.

| Parameters | Slope | | | |
|---|---|---|---|---|
| | 1 | 2 | 3 | 4 |
| Lithology | Ignimbrite | Ignimbrite | Ignimbrite | Ignimbrite |
| Slope height (m) | 10.80 | 23.70 | 8.00 | 11.15 |
| RQD (%) | 93 | 94 | 97 | 85 |
| $J_n$ | 12 | 12 | 12 | 12 |
| $J_r$ | 3 | 3 | 3 | 3 |
| $J_a$ | 2 | 2 | 2 | 2 |
| O Factor | 0.75 | 0.50 | 1.00 | 0.50 |
| $J_w$ | 0.50 | 0.50 | 0.50 | 0.50 |
| $SRF_{slope}$ | 2.50 | 4 | 2.50 | 4 |
| Q-Slope | 1.74 | 0.73 | 2.43 | 0.66 |
| β | 69.80 | 62.30 | 72.70 | 61.40 |
| Slope angle | 79 | 61 | 57 | 57 |
| Stability | Unstable | Stable | Stable | Stable |

*3.3. Rockfall Risk Rating System (RHRSmod)*

For the development of the Rockfall Risk Rating System (RHRS), the modified RHRS method [23] was selected. This method was chosen because it incorporates the slope mass rating (SMR) within its evaluation categories, whose data was previously obtained. It is based on the characterization of the rock mass, providing reliable data concerning slope behavior. Scores for the modified RHRS were obtained and are presented below (Table 7):

**Table 7.** Slope scores with RHRSmod.

| Slope | Score |
|---|---|
| 1 | 242.80 |
| 2 | 208.90 |
| 3 | 205.40 |
| 4 | 249.20 |

These scores indicate that corrective action is required for the risk of rockfall on slopes 1 and 4. As for slopes 2 and 3, continuous monitoring is recommended at these sites (Table 8).

**Table 8.** Values of RHRSmod for the slope.

| Parameters | Slope | | | | | | | |
|---|---|---|---|---|---|---|---|---|
| | 1 | | 2 | | 3 | | 4 | |
| Site | Value | Score | Value | Score | Value | Score | Value | Score |
| Slope height (m) | 23.00 | 27.00 | 11.00 | 6.00 | 17.00 | 13.00 | 15.00 | 9.00 |
| Trench effectiveness | Limited catchment | 27.00 | Limited catchment | 27.00 | Moderate catchment | 9.00 | Limited and moderate catchment | 50.00 |
| Average vehicle risk (AVR) | 27.63 | 3.00 | 14.17 | 3.00 | 21.25 | 3.00 | 31.88 | 3.00 |
| Decision sight distance (%Da) | 28.12 | 81.00 | 44.37 | 70.20 | 31.25 | 81.00 | 31.25 | 81.00 |
| Road width (Lc) meters | 5.25 | 81.00 | 5.25 | 81.00 | 5.25 | 81.00 | 5.25 | 81.00 |
| Slope mass rating (SMR) | 58.30 | 5.80 | 43.00 | 9.70 | 64.25 | 6.40 | 32.00 | 7.20 |
| Block size | 0.50 | 9.00 | 0.30 | 3.00 | 0.26 | 3.00 | 0.30 | 3.00 |
| Annual rainfall (h) mm/year | 600 | 6.00 | 600 | 6.00 | 600 | 6.00 | 600 | 6.00 |
| Rockfall frequency | Few falls | 3.00 | Few falls | 3.00 | Few falls | 3.00 | Occasional falls | 9.00 |

For Slope 1, there are rock fragments at the base of the slope, which indicates a similarity with the software's results, since the trajectory aligns with what was observed in reality (Figure 5), even though some of these fragments have been displaced during road cleaning.

For Slope 2, field observations revealed fragments similar to those shown in the image, displaying a trajectory very similar to the RocFall simulation results. For Slope 3, there is a discrepancy with the field observations, since fewer fallen fragments were noted. Moreover, the trajectory in the simulation indicates a considerable distance from the base of the slope. For Slope 4, the fallen fragments align with the trajectory indicated in the program. The Rocfall software simulates rock trajectories using a rebound coefficient depending on the nature of the slope.

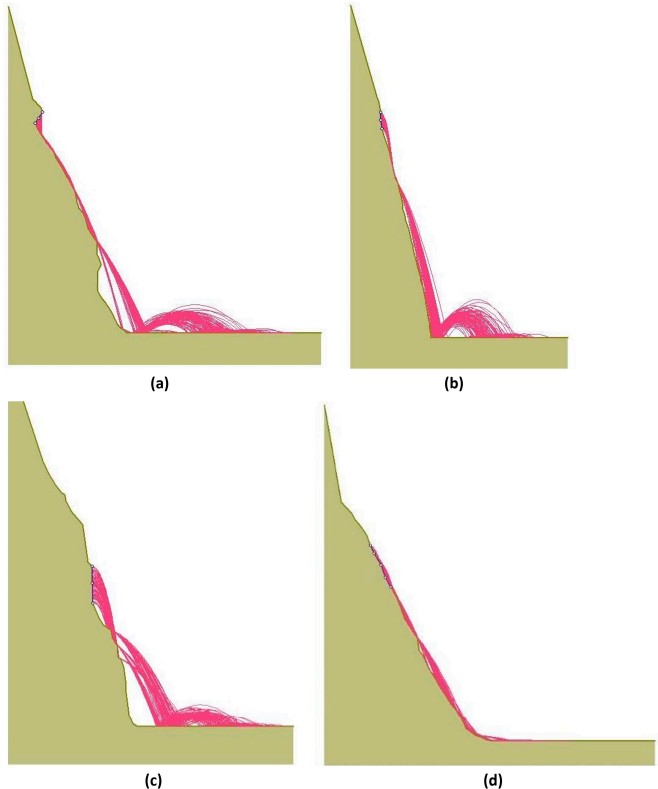

**Figure 5.** Analysis of the four slopes using Rocfall software: (**a**) Slope 1; (**b**) Slope 2; (**c**) Slope 3; (**d**) Slope 4.

### 3.4. Kinematic Analysis

By entering the data for the discontinuities and each slope into the Dips program, as outlined in the methodology, we can visually identify the poles of the discontinuities within a zone of instability. According to the method, the poles of the discontinuities that are in this zone are considered unstable. In Figure 6, we can observe the results indicating that slopes 2 and 4 are considered unstable (Table 9).

**Table 9.** Kinematic assessment of the study rock slopes.

| Parameters | Slope | | | |
|---|---|---|---|---|
| | 1 | 2 | 3 | 4 |
| Slope orientation DipDir/Dip | 087/79 | 139/61 | 119/57 | 115/58 |
| Slope height (m) | 10.80 | 23.70 | 8.00 | 11.15 |
| Joint 1 DipDir/Dip | 057/87 | 320/86 | 097/064 | 292/88 |
| Joint 2 DipDir/Dip | 036/53 | 072/70 | 194/83 | 322/86 |
| Joint 3 DipDir/Dip | N.I. | 176/86 | 354/77 | 052/86 |
| $J_r/J_a \times J_w$ | $3/2 \times 0.5 = 0.75$ | $3/2 \times 0.5 = 0.75$ | $3/2 \times 0.5 = 0.75$ | $3/2 \times 0.5 = 0.75$ |
| Frictional componnet FC (degrees) | 48 | 37 | 37 | 37 |
| Failure mode and critical joints | Stable, no failure modes | Flexural toppling with J1 | No strict failures. Stable | Flexural toppling with J1 |
| Factor of safety FS | 0.27 | >1 | >2 | >3 |

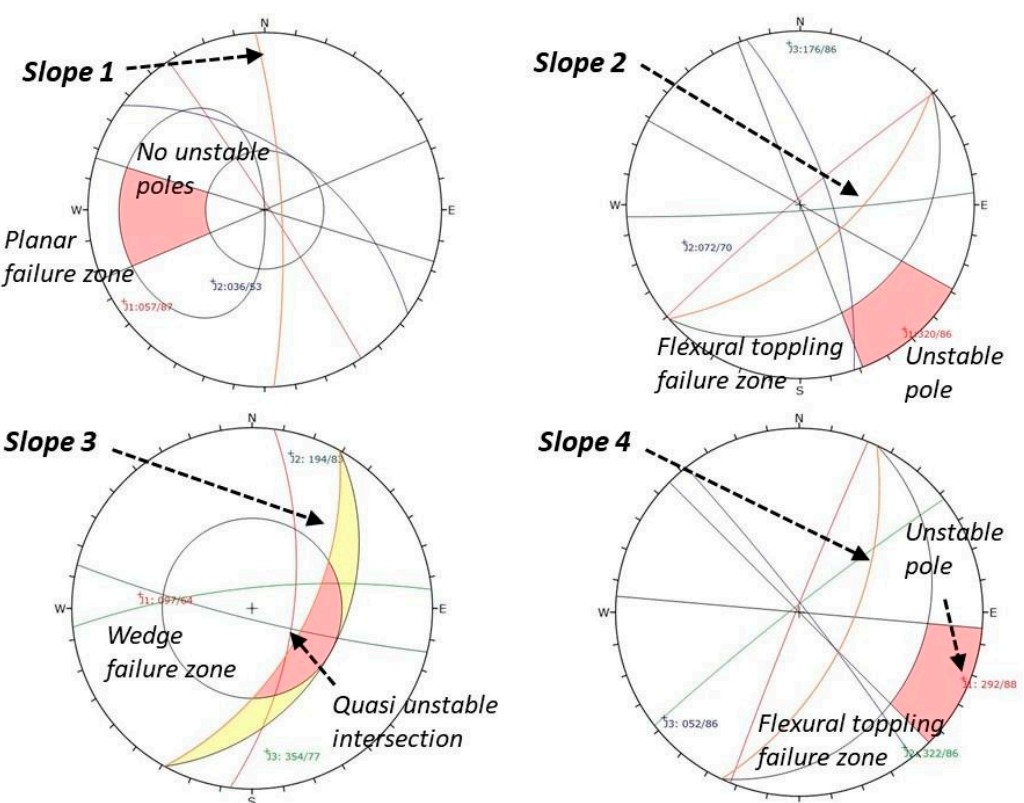

**Figure 6.** Kinematic analysis results.

## 4. Discussion

In this work, fieldwork was conducted to acquire the geotechnical parameters of the rock mass and discontinuities, alongside the creation of a photogrammetric model. Given challenges in accessing the slopes, a decision was made to integrate field analysis with remote techniques. For the slopes under examination, it was very important to compare discontinuity measurements with the results the computer programs utilizing point cloud data and automatic identification of discontinuities (DSE). This enabled the verification of field measurements with minimal adjustments, addressing errors that typically arise in the field due to the challenges of making measurements, particularly in terms of space and accessibility limitations.

Table 10, below, summarizes the results obtained for each method in assessing the stability of the studied slopes.

**Table 10.** Summary of results by the different methods.

| Applied Method | Slope | | | |
| --- | --- | --- | --- | --- |
| | 1 | 2 | 3 | 4 |
| Slope mass rating (SMR) | Partially Stable | Partially Stable | Stable | Unstable |
| Q-Slope | Unstable | Stable | Stable | Stable |
| Kinematic analysis | Stable | Stable | Stable | Stable |
| RHRSmod | Medium Risk | Low Risk | Low Risk | Medium Risk |
| Visual | Stable | Partially Stable | Stable | Stable |

Table 11 illustrates the discrepancies in input parameters for each of the criteria. It is important to note certain criteria necessitate a preliminary on-site assessment of stability as well as the favorable or unfavorable orientation of the discontinuities concerning the excavation.

**Table 11.** Summary of the key required inputs for each method.

| Applied Method | Require Parameters (Yes/No) | | | | Preliminary Stability Approach |
| --- | --- | --- | --- | --- | --- |
| | Orientation of Discontinuities | Slope Orientation | Slope Heigh | Rock MassQuality | |
| Slope mass rating (SMR) | Yes | Yes | No | Yes | No |
| Q-Slope | No | No | No | Yes | Yes |
| Kinematic analysis | Yes | Yes | No | No | No |
| RHRSmod | No | No | Yes | Yes | No |
| Visual | Yes (visual) | Yes (visual) | Yes | No | Yes |

In the case of Slope 1, although no stability issues were identified in the field, it was observed that empirical methods yielded unfavorable results in terms of stability. According to the SMR classification, Slope 1 was deemed partially stable. The Q-slope method indicates that the slope is unstable due to the fact that the slope inclination angle exceeds the threshold indicated by the method for stability under its conditions. In the kinematic analysis, the slope is a stable slope despite having a safety factor of 0.27. This discrepancy arises because there is a specific zone of the slope prone to failure, while the rest of the slope is stable. This inconsistency between the safety factor result and overall slope stability underscores the need for a nuanced understanding of the slope's stability. However, with the results of the empirical methods, the orientations of the discontinuities present a risk of failure. This aspect must be considered since, over time or during specific events, it could pose a danger to road users. In this case, Joint 2 is identified as the potential source of this problem.

In the case of the three remaining slopes, different results are observed when comparing the SMR and the Q-slope. It is important to note that the Q-slope method was created for newly excavated slopes where the inclination angle of each slope determines its stability. In this case, where the road slopes were cut by blasting, the Q-slope method provides

more accurate guidance. Slopes 2, 3 and 4 are considered stable based on their inclination angles. This methodological comparison highlights the significance of analyzing rock slopes through various methods, as each case is unique, and results should be interpreted in accordance with the specific geological and structural characteristics.

In the kinematic analysis, the stability results for the three remaining slopes were favorable, with safety factors greater than 1. In the case of Slope 4, this factor was greater than 3. These findings indicate that, despite observed evidence of small rock fragment detachment, the slopes, in general, are stable under this method. The failures or falls of these blocks may be due to localized failures.

Regarding the scores obtained in the RHRS mod method, it was identified that slopes 1 and 4 posed a greater danger in terms of rockfall detachment, with values greater than 240 points. Corrective action is deemed necessary for these slopes, and their risk should be prioritized. As for slopes 2 and 3, with scores below 210 points, ongoing observation is recommended, with mitigation works undertaken when necessary.

## 5. Conclusions

The methodology employed in this work shows the feasibility of conducting rapid, cost-effective assessments of slopes using remote and easily accessible techniques. In this instance, photogrammetry with control points (SfM) was performed. This technique provided the necessary properties to perform the analysis by the different methods proposed in this work, enabling quick and straightforward analysis of rock slopes without the need for highly specialized and often difficult-to-access equipment. A quick and timely analysis of slopes can prove crucial in preventing accidents and minimizing disruptions for road users.

The comparison of methods highlights the importance of combining results for accurate interpretation. For example, Q-slope does not consider the orientation of the discontinuities and solely considers the degree of inclination of a slope. The kinematic method and SMR consider these orientations, making them more reliable in conjunction with visual observations in the field to identify potential hazards and slope failures.

The RHRS mod is a fast and economical method relying on field-collected data and site mapping. These data encompass the physical features of the slope and the adjacent road, climatic conditions, and the geological composition. This approach allows for a comprehensive evaluation of road risk, facilitating the identification of critical areas prone to rockfall issues on the slopes.

**Author Contributions:** Conceptualization, L.J.-B., M.M., L.G.A. and R.C.V.; methodology: L.J.-B., M.M., L.G.A., R.C.V., G.B., M.M.-M., E.L. and D.G.; validation L.J.-B., M.M., E.L. and D.G.; formal analysis, L.G.A., R.C.V., G.B. and M.M.-M.; investigation: L.J.-B., M.M., L.G.A., R.C.V., G.B., M.M.-M., E.L. and D.G.; resources, L.J.-B., M.M. and D.G.; data curation: L.G.A., R.C.V., G.B., M.M.-M. and E.L.; writing—original draft preparation, L.J.-B., M.M., L.G.A., R.C.V., G.B. and M.M.-M.; writing—review and editing, L.J.-B., M.M., G.B., M.M.-M., E.L. and D.G.; supervision, L.J.-B., M.M., E.L. and D.G.; project administration: L.J.-B., M.M. and D.G.; funding acquisition: M.M. All authors have read and agreed to the published version of the manuscript.

**Funding:** This research was partially funded through the Geotechnic Master's Degree of the Faculty of Engineering in Earth Science (FICT (acronym in Spanish)) of the Polytechnic University (ESPOL—Guayaquil—Ecuador) through a scholarship.

**Data Availability Statement:** The data presented in this study are available on request from the corresponding author.

**Acknowledgments:** This paper is part of a final thesis for the ESPOL Geotechnics Master.

**Conflicts of Interest:** The authors declare no conflict of interest.

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
