# Peer review of "An Analysis of Slope Stability in the Penipe–Baños Road by Applying Empirical Methods, Kinematic Analysis and Remote Photogrammetry Techniques"

_geosciences, doi:10.3390/geosciences13120366_

Round 1

Reviewer 1 Report

Comments and Suggestions for Authors

Dear authors,

the paper is an interesting case study using different approaches to evaluate the slope stability of some cuts in a new road, in order to compare them. However, the scope, advantages and disadvantages of each approach is not clear enough. In the discussion and conclusions I would add some comments about that.

In order to facilitate the reading I would include a table with the different approaches and the main data used (direction and dip direction of the discontinuities, orientation and height of the slope, and so on) for the different approaches. 

Regarding the SfM it is not clear the degree of similarity of the measures obtained with the photogrammetry to those obtained through geomechanical stations. Just in case they were different it would be necessary to explain the used criteria for chosing the direction and dip direction of the different discontinuities.

In the attached pdf you will find some more comments. Some mistakes have been highlighted with no associated comments.

Author Response

REPLY TO REVIEWER 1

  • Line 1

The paper is an interesting case study using different approaches to evaluate the slope stability of some cuts in a new road, in order to compare them. However, the scope, advantages and disadvantages of each approach is not clear enough. In the discussion and conclusions, I would add some comments about that. In order to facilitate the reading. I would include a table with the different approaches and the main data used (direction and dip direction of the discontinuities, orientation, and height of the slope, and so on) for the different approaches. Regarding the SfM it is not clear the degree of similarity of the measures obtained with the photogrammetry to those obtained through geomechanical stations. Just in case they were different it would be necessary to explain the used criteria for chosing the direction and dip direction of the different discontinuities. In the attached pdf you will find some more comments. Some mistakes have been highlighted with no associated comments.

Reply:

Thank you very much for the effort of reviewing the work, we believe that your contributions have greatly improved it. Below these lines we explain one by one each of the changes made.

We have included a new table (table 11) that summarizes the main input data required for each method. Regarding the orientations of discontinuities and the heigh of the slope it is indicated in each method if that information is needed (e.g. table 8 and table 9)

  • Line 2 – 3

I would delete some commas in the title.

Reply: Done (only one) the rest is necessary

  • Line 19

Was

Reply: Done

  • Line 30

I would consider to use “different” instead of “the”.

Reply: Done

  • Line 42

Could you explain why those slopes and the material that outcrop in the slopes (pyroclastic materials or lavas)?

Reply: We choose different kind of rock failures

  • Line 56

It would be better to use "geomechanical" as below.

Reply: Done

  • Line 66 – 67

Probably it is better to use rock matrix and rocky massif as you did in 2.2

Reply: Done

  • Line 74

Rockfall Hazard Rating System

Reply: Done

  • Line 75 – 76

It seems that you refer to SMR, Q-Slope and RHRS, but you refer to RHRS.

Reply: This comment only applied to RHRS, we add some word to clarify

  • Line 79 – Figure 2

Are you using the same vertical and horizontal scale?

Reply: The scale bar indicates the real scale of the photo.

  • Line 83

I miss some references.

Reply: references 7,8,9 and 10 can be included here

  • Line 101

I suppose it is a Brunton 9077.

Reply: Yes, we use a Brunton compass.

  • Line 102

Is it the same BOSCH GLM400C used in the next section?

Reply: Yes, it is.

  • Line 116

Do you mean SfM?

Reply: Yes.

  • Line 120

You mentioned the models for the compass, the laser meter,... so I would mention the model for the other materials that you have used.

Reply: A Schmidt original sclerometer

  • Line 142

Maybe it is better to use a . or ;

Reply: Done

  • Line 143

I would include (Jw).

Reply: Jw is only for the tunnel Q index, not slopes

  • Line 143

I consider that you should explain the relationship between SRFa and SRFb.

Reply: We refer the reader to reference 17

  • Line 149 – Table 3

What it this?

Reply: For the application of the SRF the high score among SRFa, SRFb and SRFc should be applied

  • Line 152 – Equation 2

Probably something is missing.

Reply: The + must be elliminated

  • Line 161

I would consider this section as 2.6 because it is using some data not obtained from geomechanical stations.

Reply: We believe that we should leave it. Main information is obtained from geomechanical stations.

  • Line 164

I consider it is better to use excavated.

Reply: Done

  • Line 180

In my opinion this subsection is not necessary. Just in case you consider it is, I would title as modified RHRS.

Reply: We agree with the observation. Is better delete this title.

  • Line 185

A reference would be appreciated.

Reply: included: (Budetta, 2004)

  • Line 190

I miss a paragraph about Rockfall software.

Reply: included

  • Line 191

I would consider this section 2.5

Reply: This can change the meaning of the document

  • Line 211

I think that you mean “they”.

Reply: Done

  • Line 216

3a and 3c seem the same slope. I would include a vertical and horizontal scale and the meaning of the colored scale placed in the right side of each picture.

Reply: They are not the same slope.

  • Line 219

I would include (SMR)

Reply: Done

  • Line 227

I suppose that P means Planar and T means Toppling. Are you using the Romana (1985) ratings?

Reply: Yes

  • Line 330
    Could you complete this sentences?
    Reply: We forgot to erase these words
  • Line 376 – References 10
    Journal is missing
    Reply: Done
  • Line 388 – References 18
    Probable this reference is not correct
    Reply: Done

Reviewer 2 Report

Comments and Suggestions for Authors

Having read your paper, I find it quite interesting, and it makes an important contribution to the field of road slope safety. However, I would suggest 1) Further development of the introduction with a better literature review, 2) Better description of the methodology used and 3) Highlight photogrammetric methods as an alternative method for the analysis of dangerous slopes, maybe you could also mention the use of modern technology such as drones etc.

In the attached document there are comments on points that I think need attention.

Author Response

REPLY TO REVIEWER 2

  • Line 1-3

General comment of review 2:
Having read your paper, I find it quite interesting, and it makes an important contribution to the field of road slope safety. However, I would suggest

1) Further development of the introduction with a better literature review,

2) Better description of the methodology used and

3) Highlight photogrammetric methods as an alternative method for the analysis of dangerous slopes, maybe you could also mention the use of modern technology such as drones etc.
In the attached document there are comments on points that I think need attention.

Reply:

Thank you very much for the effort in reviewing and improving our research. We have improved both the introduction and methodology explanations. In this work we have not used drones, we have tried a low-cost manual methodology. Below these lines we indicate all the changes we have made following the reviewer's specific instructions

  • Line 51
    Rephrase: No global nor cyclical failure were observed in the area so it was considered that no further analysis was required.
    Reply: Done
  • Line 131
    Consider to add the reference: Romana, M., Tomás, R., & Serón, J. B. (2015). Slope Mass Rating (SMR) geomechanics classification: Thirty years review. 13th ISRM International Congress of Rock Mechanics, 2015-MAY.
    Reply: We change the refence using this reference

Delete numbering
Reply: Done

  • Line 180
    It is wrong to separate in subsection while 2.5.2 not exists! Also the title is wrong, probably you forgot to erase it?
    Reply: We are agree with the observation. Is better delete this title
  • Line 212
    Describe more analytical the below figure. What means the red, green and blue colors. Also the colorbar numbering and title it is not readable.
    Reply: Colors indicate fracture orientations
  • Table 5RQD(%)
    How it is measured? From photoes I have the impression of smaller values.
    Reply: RQD has been obtained using tthe joints per meter criteria
  • Table 6 – Stability
    Different conclusion flom SRM.
    Reply: It is discussed later
  • Line 246
    Better not change paragraph for this sentences
    Reply: Done
  •  
  •  

Round 2

Reviewer 2 Report

Comments and Suggestions for Authors

I have no more comments to make, just an observation about figure 3, the numbers in the color bar are not visible and the zeros should be cut if possible because they give the impression that they are 3000000 instead of 3. Also you can circle the 3 families of joints in the figure.

Author Response

Dear reviewer,

Thanks you very much for your insights

we have modified figure 3 according to your comments

regards

The authors